# Molecular Characterization of Extended Spectrum β-Lactamase (ESBL) and Virulence Gene-Factors in Uropathogenic *Escherichia coli* (UPEC) in Children in Duhok City, Kurdistan Region, Iraq

**DOI:** 10.3390/antibiotics11091246

**Published:** 2022-09-14

**Authors:** Salwa Muhsin Hasan, Khalid S. Ibrahim

**Affiliations:** 1Department of Medical Lab Technology, College of Health and Medical Technology/Shekhan, Duhok Polytechnic University, Duhok 42001, Kurdistan Region, Iraq; 2Department of Biology, Faculty of Sciences, University of Zakho, Zakho 42002, Kurdistan Region, Iraq

**Keywords:** *E. coli*, ESBL, UTIs, children, Duhok, Iraq

## Abstract

Background: The presence of extended-spectrum β-lactamase (ESBL)-producing bacteria among uropathogens is significantly increasing in children all over the world. Thus, this research was conducted to investigate the prevalence of *E. coli* and their antimicrobial susceptibility pattern, and both genes of ESBL-producing *E. coli* resistant and virulence factor in UTIs patients among children in Duhok Province, Kurdistan, Iraq. Method: a total of 67 *E. coli* were identified from 260 urine samples of pediatric patients diagnosed with UTIs aged (0–15 years) which were collected from Heevi Pediatric Teaching Hospital, from August 2021 to the end of February 2022. Result: a high proportion of UPEC infections at ages <5 years and the rates among girls (88%) were significantly higher than those among the boys. A wide variety of *E. coli* are resistant to most antibiotics, such as Amoxicillin, Ampicillin and Tetracycline, and 64% of them were positive for ESBL. Interestingly, the presence of both the ESBL marker genes (*bla_TEM_,* and *bla_CTX-M_*) as well as both virulence marker genes (*pai* and *hly*) were detected in above 90% of *E. coli*. Conclusion: the data illustrate an alarming increase in UPEC with ESBL production and the emergence of multidrug-resistant drugs in the early age of children. The public health sectors should further monitor the guidelines of using antibiotics in Kurdistan, Iraq.

## 1. Introduction

Urinary tract infections (UTIs) are the most common severe bacterial infections seen in clinical settings worldwide [1,2]. It is estimated that annually there are over 150 million occurrences of UTIs, which are medically expensive [3,4]. On yearly basis, the total healthcare cost of UTIs in the USA, whether for its management or treatment, is over 3.5 billion USD [5], and around 630 million USD for pediatrics UTIs [6]. Pediatric UTIs are estimated to impact 2.4 to 3 % of all American children each year, resulting in more than 1.5 million doctor visits [7], of which 5–14% are dedicated to emergency visits [8]. The incidence of UTIs is common during childhood, and affects in both boys and girls equally during the first year of life [9]. UTIs are considered the second most frequent bacterial infection in children throughout the first seven years of life, and girls account for the majority of infections, approximately 8.4% and less than 1.7% for boys [10,11,12].

It is known that *Escherichia coli* (*E. coli*) is the most prevalent bacterial pathogen that causes UTIs and is responsible for 80% of UTIs that occur in children [13]. Numerous virulence factors, such as adhesins, toxins, and siderophores, are present in uropathogenic *Escherichia coli* (UPEC) strains and contribute to the spread of infection [14,15]. The ability of bacteria to adhere to the host cells in the urinary system and prevent quick clearance by the bulk flow of urine is one of the most crucial virulence factors [16]. The most significant virulence component released by uropathogenic *E. coli* is a lipoprotein known as α-hemolysin (*hly*A), which is connected to higher UTIs that result in pyelonephritis [17]. In addition, *E. coli* that has colonized the body may produce toxins that create an inflammatory reaction, which is one potential cause of the symptoms of UTIs [18].

The majority of enzymes that can break down penicillins, third-generation cephalosporins, and monobactams are known as extended-spectrum β-lactamases (ESBLs) [19]. Because antibiotics hydrolyzed the β-lactam ring and bound covalently to their carbonyl moiety, they were ineffective as a therapy [20]. There are very few antibiotic choices for treating a bacterium that generates ESBL. For serious infections caused by ESBL-producing organisms, carbapenems are the drug of choice [21]. However, studies reported that carbapenem-resistant isolates do exist [21,22]. 

The prevalence of ESBL in Enterobacteriaceae with UTIs patients is increasing and has become a significant cause of morbidity and a major problem all over the world [23]. The World Health Organization declared in 2021 [24] that the category of Enterobacteriaceae that produce ESBL offers the greatest danger of infection to the health of the global population. One of the most prevalent bacteria carrying ESBL genes is *E. coli*, which belongs to the Enterobacteriaceae family. Due to its rapid growth, this bacterium represents a threat to both sexes in the community at different phases of life [25,26].

It has been reported that the prevalence of *E. coli* with ESBL in UTIs varies greatly between different nations [27,28,29,30,31]. In the Kurdistan Region, Iraq, several studies have been conducted on the prevalence of UTIs among adults [32,33,34] and reporting the antibiotics susceptibility test only in children [35,36,37]. Therefore, this study is undertaken to identify ESBL-producing *E. coli*, to detect their ESBL resistant marker genes and five virulence factors marker genes, and to determine their antimicrobial susceptibility pattern among children in Duhok Province, Kurdistan Region, Iraq.

## 2. Results

### 2.1. E. coli Isolation with Phenotypic and Genotypic Detection

In this current study, a total of 260 out- and in-patient children (<15 years) were recruited, from August 2021 and February 2022. From these cultures, 150 (57.7%) were phylogenetic positive bacterial cultures. From 150 samples, *E. coli* was the predominant pathogenic bacteria, and 67 (44.7%) of them were confirmed phenotypically and molecularly by the species-specific gene (*uidA*). Phenotypically, *E. coli* was identified on MacConkey and EMB agars as well as Gram stain and biochemical tests (Oxidas, IMViC, TSI). In addition, all isolated *E. coli* were genotypically confirmed by the species-specific gene (*uidA*) (Figure 1a).

### 2.2. Antimicrobial Susceptibility Pattern

All the isolated *E. coli* underwent an antimicrobial susceptibility test, and the results of the 19 antibiotics and their percentage (yellow colors) are shown in Figure 2. Isolated *E. coli* were resistant to Tetracycline around (97%) and approximately 85% of them were highly resistant to Amoxicillin and Ampicillin. Furthermore, around 70% of isolated *E. coli* were resistant to Nalidixic acid, Cefotaxime, Cefixime, Trimethoprim/ Sulfamethoxazole, Gentamicin and Ceftriaxone. In addition, around (57%) of these *E. co**li* were resistant to Amikacin, (51%) to Ciprofloxacin and (45%) to Aztreonam. In contrast, over 85% of *E. coli* were highly susceptible to Chloramphenicol, Nitrofurantoin and Imipenem.

### 2.3. Phenotypic and Molecular Detection of ESBL Production

All *E. coli* were subjected to detect the presence of the production of ESBLs by the double disc synergy test (DDST), shown in Figure 2 and Appendix A. The results show that 43 (64%) of *E. coli* were positive for ESBLs production by DDST. They were also tested for whether they were harboring ESBL genes or not by PCR amplifications as shown in Figure 1. Figure 2 presents individual and collective detecting of marker genes. The fragments which produced ESBL genes include TEM, CTX-M, and SHV, confirmed by agarose gels (Figure 1). Among the ESBL marker genes, TEM and CTX-M were more abundant 63 (94%) and 60 (90%), respectively, and the SHV had the lowest abundance of 1 (1.4%). It is interesting to note that only 3 out of the already-decided 24 non-producer ESBLs *E. coli* turned out to be non-ESBL, the rest were ESBL (Figure 2). Table 1 illustrates the presence of these ESBL marker genes in all of the isolated *E. coli* strains and 58 of the isolated strains harboring both TEM and CTX-M genes. Furthermore, the TEM gene was found in the four of the *E. coli* strains while only one strain harbored three TEM, CTX-M, SHV genes and another strain has only the CTX-M gene.

### 2.4. Molecular Detection of Virulence Factors Genes 

The isolated *E. coli* strains were also tested for the presence of the five virulence factor genes and their PCR product size on agarose gels, and these are shown in Figure 1. The total number and prevalence rate of the five virulence factor genes in *E. coli* isolates are also shown in Figure 2 and Appendix A. Of the 67 *E. coli* isolates, 63 (94%), 55 (82%), 27 (40%) 9 (13%) and 4 (6%) of isolates possess pathogenicity island (*pai*), hemolysin (*hly*), a fimbrial adhesion (*afa*), S-fimbrial adhesion (*sfa*), and cytotoxic necrotizing factor-1 (*cnf-1*), respectively. Table 1 illustrates the details of virulence factor genes that existed in all of the isolated *E. coli* strains. The main point of interest is that 21 of the isolated *E. coli* strains had both *pai* and *hly* genes, and 18 strains had three genes, *pai*, *hly* and *afa*. Furthermore, five strains possess three genes, *pai, hly,* and *sfa* and, the other five strains possess both *pai,* and *afa*. Moreover, two of these *E. coli* strains possess *pai, hly, afa* and *sfa*, one possesses *pai*, *hly*, *sfa* and *cnf-1*, two possess *hly* and *afa*, three possess *pai*, *hly* and *cnf-1*, one possesses *pai* and *sfa*, five possess *pai*, and one possesses *hly and afa*.

### 2.5. Relationship between UTIs by E. coli and Children Age

The total number of children with UTIs is overwhelming, at earlier ages than the pre- and school aged children. The Spearman’s correlation coefficient indicates that there was a significant negative correlation between the UTIs with UPEC and age of children; (r = −0.830, *p*-value = 0.0003) (Figure 3).

### 2.6. Relationship between Antibiotic Resistance and Marker Genes of E. coli and Children Ages Having UTIs

Figure 4 illustrates the total number of antibiotics resisted and the marker genes (i.e., antibiotics resistance and virulence factors) of *E. coli*, in pediatric patients of all ages. It highlights the fact that they are significantly higher in early ages than pre- and school age groups. Figure 4a shows the Spearman’s correlation coefficient, indicating that there was a significant negative correlation between the children of all ages and UPEC resistant to antibiotics. The findings of the significant relationships are as follows: Tetracycline (TE; r = −0.8156, *p*-value = 0.0004), Amoxicillin (AX; r = −0.8266, *p*-value = 0.0003), Ampicillin (AM; r = −0.8266, *p*-value = 0.0003), Nalidixic acid (NA; r = −0.5399, *p*-value = 0.0404), Nitrofurantoin (F; r = −0.5399, *p*-value = 0.0404), Cefotaxime (CTX; r = −0.665, *p*-value = 0.0084), Cefixime (CFM; r = −0.7456, *p*-value = 0.002), Trimethoprim/Sulphamethoxazole (SXT; r = −0.629, *p*-value = 0.0138), Amoxicillin/Clavulanic acid (AMC; r = −0.7961, *p*-value = 0.0007), Ceftazidime (CAZ; r = −0.6443, *p*-value = 0.0112), Ceftriaxone (CRO; r = −0.5617, *p*-value = 0.0316), Amikacin (AK; r = −0.63, *p*-value = 0.0137), Aztreonam (ATM; r = −0.6228, *p*-value = 0.0151), Chloramphenicol (C; r = −0.5838, *p*-value = 0.0251), and F (r = −0.8295, *p*-value = 0.0003). In addition, a similar trend (although statistically not so significant) was observed in the other antibiotics; Gentamicin (CN; r = −0.3393, *p*-value = 0.2145), Ciprofloxacin (CIP; r = −0.4155, *p*-value = 0.1242), Levofloxacin (LEV; r = −0.2894, *p*-value = 0.2924), Norfloxacin (NOR; r = −0.2337, *p*-value = 0.398) and Imipenem (IPM; r = −0.3334, *p*-value = 0.2239). In addition, there was a significant negative correlation between the ESBL-producing *E. coli* and the age of children (ESBL; r = −0.5624, *p*-value = 0.0313) (Figure 3).

Furthermore, the Spearman’s correlation coefficient which indicated there was a significant negative correlation between the children’s age and both marker genes of ESBL-producing and virulence factors of UPEC infection (Figure 4b). The findings of the significant relationship are as follows: TEM (r = −0.8189, *p*-value = 0.0003), CTX (r = −0.7594, *p*-value = 0.0016), *pai* (r = −0.7986, *p*-value = 0.0006) and *hly* (r = −0.7311, *p*-value = 0.0028). In addition, a similar trend (although statistically not so significant) was observed in the other marker genes, SHV (r = −0.1237, *p*-value = 0.8), *afa* (r = −0.4368, *p*-value = 0.1077), *sfa* (r = −0.04953, *p*-value = 0.8617) and *cnf-1* (r = −0.5234, *p*-value = 0.0557).

### 2.7. Relationship between Antibiotics Resistance and Virulence Factors Genes

Figure 5 illustrates the relationship between the antibiotic-resistant genes and virulence factors of UPEC infection, using Spearman’s correlation coefficient. There were highly significant positive correlations between TEM and CTX (r = 0.98, *p*-value < 0.0001), *pai* (r = 0.96, *p*-value < 0.0001) and *hly* (r = 0.94, *p*-value < 0.0001), and also between CTX, and *pai* (r = 0.96, *p*-value < 0.0001) and *hly* (r = 0.95, *p*-value < 0.0001). Furthermore, despite the very low prevalence of *cnf-1*, it was also found that *cnf-1* was a significant positive correlation with TEM (r = 0.60, *p*-value = 0.023), CTX (r = 0.60, *p*-value = 0.021), *pai* (r = 0.63, *p*-value = 0.016), and *hly* (r = 0.62, *p*-value = 0.018). 

## 3. Discussion

The incidence of pediatric UTIs has increased dramatically in recent years worldwide, which is caused by a variety of pathogens, particularly Gram-negative bacteria. This study revealed that *E. coli* was the predominant uropathogen in children in our community, where the prevalence of the infected girls was higher than in boys. These findings were in agreement with the recent results obtained by several studies in Iraq, in Duhok city [35], Erbil city [36,37], and Baghdad city [38], as well as a Child Cohort Study in Taiwan with UTIs between 2004 and 2018 [39]. In fact, there are many factors that can affect the interpretation of UTIs in girls, particularly *E. coli*. Among these factors are the anatomical structure of the urogenital system, and girls have a shorter distance between the urethral and anal opening [40]. It has been previously suggested that there is a relationship between the arising of the dominant fecal flora, coliform bacteria, colonizing the perineum which enters and ascends to the urinary tract in girls [41,42,43]. In fact, studies have demonstrated that the UPEC strains possess specific properties. For instance, both pili and fimbriae can mediate bacterial attachment and invasion to the uroepithelial cell surface, allowing them to overcome the host defenses [16,42]. In addition, the majority of pathogenic bacteria in UTIs are the commensal bacteria and their pathogenicity might result from microbiota dysbiosis [42].

In this current study, the frequency of UTIs by *E. coli* was widely reduced in older children. It showed high infection rates in children aged between 0–3 years and during toilet training for 4–5 years, particularly in girls. The increase in the incidence rate of UTIs at these ages may be due to the use of diapers and the wet tissues for infants and during toilet training for 4–5 years [44]. Boys with UTIs have rarely been detected in this study and this is due to the fact that almost all boys have undergone circumcision in their first week after birth [45]. Thus, this finding is dissimilar to Western studies and highlights the notion that uncircumcision in boys is the main reason for the higher UTIs proportion than girls, which is inverse in our communities [46,47].

In this study, the antimicrobial susceptibility testing of *E. coli* showed that 97% of them were highly resistant to Tetracycline. This finding is higher than previous studies conducted with different child age groups in Iraq: 63% [48], 83% [33] and 80% [49]. High resistance may primarily be attributed to the diversity of genes in various bacteria that regulate the efflux of Tetracyclines from the cell [50]. Furthermore, this study demonstrated around 85% higher resistance to Penicillin agents, including Amoxicillin and Ampicillin. This may be due to the misuse of older antimicrobials that have made it a less acceptable choice, such as Penicillins agent [51]. Although it has been suggested that certain Penicillins, when taken with β-lactamase inhibitors (clavulanic acid), are useful in treating some infections caused by bacteria that produce ESBLs [50], in this study, 64% of *E. coli* were resistant to Amoxicillin-clavulanic acid. This finding is in agreement with a similar laboratory-based recent study conducted in Turkey [52], while half of it was found in a study conducted in a European country such as Poland [53].

It has been emphasized that Imipenem is considered a stable antibiotic against the enzyme ESBLs produced by Enterobacteriaceae as well as preferred for treating serious infections caused by bacteria producing β-lactamase [54]. In the current study, approximately 88% of *E. coli* were sensitive to Imipenem. This finding was slightly lower when compared with other results from studies in Iraq, roughly 95.2% in Erbil city [36,37], and 100% in Baghdad [38]. It is surprising that the very recent data estimated that *E. coli* have not been observed to be resistant to Imipenem in UTIs patients under 12 old in Iran [55], and in the various ages in Iraqi studies, such as those in Duhok city [56] and Erbil city [57], as well as in Saudi Arabia [58]. In other words, *E. coli* resistance to Imipenem is steadily increasing when compared with studies of UTIs in various age groups, and this is due to the use of over-the-counter antibiotics and the ease of finding and buying antibiotics without a prescription. The public health sector has not verified the guidelines for how to control the selling of antibiotics. In addition, both Nitrofurantoin and chloramphenicol are considered the second most effective antibiotic against these isolates, roughly 85% in this present study. A similar result was found for Nitrofurantoin in a study conducted in Ethiopia for children under 15 years [59], while a dissimilar finding for *E. coli* resistance to chloramphenicol was recorded in a study in Iran, accounting for about 72% [60].

It has been previously noted that *E. coli* which produce extended-spectrum β-lactamase (ESBL) are becoming more common in the community worldwide [61]. In this study, all isolates of *E. coli* were subjected to detect ESBLs phenotypically in a clinical microbiology laboratory and approximately 64% of *E. coli* were positive for ESBLs production. The prevalence rate ESBL-producing *E. coli* was higher compared with several studies from Middle Eastern countries, approximately 30% in both Qatar [27] and Iran [28], about 41% in Turkey [29], and also in European countries, about 16.8% in Sweden [31] and around 2% in Spain [30]. According to Rupp [19], the DDST is much more sensitive and specific. The DDST, however, can produce false-positive or false-negative results [22,62]. Therefore, all isolates of *E. coli* were subjected to PCR using TEM, CTX-M, and SHV-specific primers for the detection of ESBL genes. Approximately 64% of *E. coli* were positive for the DDST, while 94% and 89% of them were identified as possessing ESBL genes, TEM and CTX-M, respectively. However, a total of 19 *E. coli* strains have the ESBL marker genes from 24 non-ESBL producer *E. coli*, and they were identified by DDST. This finding suggests that the detection of ESBL by PCR is more sensitive than the DDST.

In this study, all isolated *E. coli* were confirmed by the PCR amplification of *E. coli*-specific marker gene (*uidA*). As is known, this gene is widely used in the identification of *E. coli* from clinical samples [63,64,65,66]. Evidence demonstrated that the presence of *pai* in pathogenic strains carries genes encoding more virulence factors, such as adhesins, toxins (*cnf-1* and *hly*), invasions, and iron absorption systems [67,68]. Altogether, these bacterial virulence factors enhance their ability to adhere, invade and colonize a host and increase the pathogenicity [69,70]. These virulence factors also help pathogenic bacteria to become resistant to immune defenses, such as phagocytosis, the complement system, or adaptive immune responses [69,70,71]. Altogether, these gene factors in UPEC contribute to urinary virulence associated with severe UTIs [18]. Thus, in this study, the high prevalence of virulence factors in the UPEC strain are major causative agents for UTIs in children in Iraq, with a high prevalence rate of both *pai* (94%) and *hly* (82%) in almost UPEC strains. The *hly* gene finding was similar (81%) to a very recent study of uropathogenic *E. coli* in Egyptian patients (age range 23–56) [72]. These findings were dissimilar to several recent studies among various age groups with UTIs, for *pai* virulence gene, approximately 72% higher when compared to a study in Zakho City [48] and around 20% higher in three different cities in Kurdistan [33], and the *hly* virulence gene was found to be 60% higher by [48] and 45% higher by [33]. Whereas *afa* was roughly similar when compared with a study in Duhok City.

In addition, further analysis was conducted to answer whether the virulence genes have a role in increasing antibiotic-resistant genes or not. Only one study was reported in Egypt by Abd El-Baky et al. [72] which found a significant positive correlation between CTX and *hly*, similar to this study. Thus, further molecular studies of whole genome sequencing are recommended to highlight the relationship between the virulence factors and antibiotic-resistant genes of these pathogenic *E. coli.*

Regarding the age, it is noteworthy in this study that the negative correlation between the age of pediatrics and UTIs by UPEC with their resistance to antibiotics and both genes of resistance and virulence factors. This finding has not been previously reported in Iraq. A recently reviewed study has highlighted the reduction in UTIs in children, particularly after age 6 [10]. In addition, the high prevalence of UTIs with *E. coli* in pediatrics depends on other factors as well. These factors together can cause more UTI outcomes in children than in adolescents. A study by Pärnänen et al. [73] reported that although infants were not exposed to antibiotics, their gut microbiome contains a higher abundance of antibiotic resistance genes than adults. Pärnänen and their colleagues found that the features of infants’ fecal antibiotic resistance genes and mobile genetic element profiles were inherited from their mother’s gut and breast milk microbiota.

The limitation of this study is the lack of molecular characterization, 16s rRNA gene and whole genome sequencing of *E. coli* for the phylogenetic analysis might help understand the epidemiology and clinical microbial infection of UPEC in our community.

## 4. Materials and Methods

### 4.1. Sample Collection

The current study was conducted in the microbiology Laboratory of Heevi Pediatric Teaching Hospital in Duhok Governorate, Iraq, from August 2021 to the end of February 2022. A total 260 urine samples were collected from in-patient and out-patient children 0–15 years old with clinical suspicion of UTIs and not receiving antimicrobial treatment.

### 4.2. E. coli Isolation and Identification

Collected urine samples were cultured on MacConkey agar and Blood agar (containing 5% human blood) and aerobically incubated overnight at 37 °C. A growth of a single organism with a count of ≥10^5^ colony-forming unit (CFU)/mL was considered to represent the UTIs [74,75,76]. The lactose-fermenting colonies were selected and sub-cultured on both MacConkey and Eosin methylene Blue (EMB) agar to obtain a pure culture and to identify *E. coli* by green metallic sheen on EMB and Pink colonies on MacConkey agar. Based on the morphological and biochemical characteristics of these pure colonies was identified according to study by [77].

### 4.3. Antimicrobial Susceptibility Test and ESBLs Detection

All isolated *E. coli* were subjected to antibiotic susceptibility test using Kirby–Bauer disk diffusion method by spreading the inoculated sterile swab on Muller–Hinton agar incubated overnight aerobically at 37 °C according to [78]. Nineteen antibiotics were used, supplied by (Bioanalyses, Turkey), Amikacin (AK; 10 mg), Ceftazidime (CAZ; 30 mg), Amoxicillin (AX; 25 mg), Cefotaxime (CTX; 30 mg), Nitrofurantoin (F; 100 mg), Gentamicin (CN; 10 mg), Imipenem (IPM; 10 mg), Trimethoprim/ Sulphamethoxazole (SXT; 25 mg), Tetracycline (TE; 10 mg), Ciprofloxacin (CIP; 10 mg), Levofloxacin (LEV; 5 mg), Amoxicillin/Clavulanic acid (AMC; 30 mg), Aztreonam (ATM; 30 mg), Ampicillin (AM; 25 mg), Ceftriaxone (CRO; 10 mg), Chloramphenicol (C; 10 mg), Cefixime (CFM; 5 mg), Nalidixic acid (NA; 30 mg) and Norfloxacin (NOR; 30 mg). The diameter of the inhibition zone around antibiotic disks was measured according to the Clinical and Laboratory Standards Institute [79].

A double-disk synergy test was used to identify *E. coli* ESBL producer and five antibiotic disks (Cefotaxime, Ceftazidime, Ceftriaxone and Aztreonam were placed 30 mm from an amoxicillin/clavulanate in the center) were used on Muller–Hinton agar according to the method described by [80].

### 4.4. Bacterial DNA Extraction from E. coli

Bacterial DNA was extracted from isolated *E. coli* using the Quick DNA Extraction kit (Guangzhou Dongsheng Biotech Co., Ltd.) following the manufacturer’s protocol. Bacterial DNA quality was achieved by (NanoDrop™ One UV-Vis Spectrophotometer, Thermo Fisher Scientific, Waltham, MA, USA) and then stored at −20 °C for DNA amplification. Different primers were used for amplification of these marker genes (Macrogen, Seoul, Korea) as described in Table 2. Purified DNA was used for PCR amplification of 9 genes (Table 3). A species-specific primer for *E. coli*, *uidA* gene. After confirming *E. coli*, 3 primers were used to detect ESBL genotypes, *bla_CTX_*_-M_, *bla_TEM_* and *bla_SHV_* β-lactamase genes. In addition, five primers were used for detecting the five virulence genes, including pathogenicity island (*pai*), hemolysin (*hly*), S-fimbrial adhesion (*sfa*), cytotoxic necrotizing factor-1 (*cnf-1*), and a fimbrial adhesion (*afa*).

The amplifications of DNA for each gene were carried out in PCR tubes containing master mix and consisting of 7 µL Tag master (Guangzhou Dongsheng Biotech Co., Ltd., Guangzhou, China), 1 µL of each forward and reverse primers (10 pmol/µL), 2 µL pure DNA (25–50 ng/µL), and adding 9µL of free-nuclease water to make a final volume of 20 µL. The amplification conditions of each gene are described in Table 2.

The DNA lengths of each fragment produced with these 9 primers were confirmed by running 7 µL of each PCR products on 1.2 % agarose gels in 1× TAE buffer and added 5 µL of Safe Gel stain Dye (Guangzhou Dongsheng Biotech Co., Ltd., Guangzhou, China) and the electrophoresis was performed at 85 V for 35 min. The agarose gel was visualized under the UV radiation (Cleaver Scientific Ltd., Rugby, UK). The images of DNA bands were captured, and the estimated amplicon size were compared with the 100 pb DNA ladder (Guangzhou Dongsheng Biotech Co., Ltd., Guangzhou, China) and 300 bp (GeneDirex, Inc, Taoyuan, Taiwan).

### 4.5. Statistical Analysis

GraphPad Prism 9.4.1 was used to analyze the data. Spearman’s method was used for nonparametric correlation between the children age and the frequency of UPEC, antibiotics resistance patterns and genes of both ESBL production and virulence factors. Significance was considered to be established when *p* < 0.05.

### 4.6. Ethical Approval

The approval for conducting this study was given by the Ethical Committee of Duhok Directorate General of Health (ethical code n 18082021-8-15) and the Ethical and Protocol Review Committee of the Biological Sciences Committee (BSCZ) at the University of Zakho (ID: “BSCZ/28/7/2021”).

## 5. Conclusions

In conclusion, the data illustrates that *E. coli* remain the most predominant bacteria among pediatrics with UTIs, particularly for girls. They also state an alarming increase in *E. coli* with ESBL production of UTIs in children of ages younger than the pre- and school age group of children. The *E. coli* ESBL-producer was more resistant to antibiotics than the non-producer. Moreover, Imipenem and Nitrofurantoin are considered to be the most effective antibiotic choices for the treatment of pediatric UPEC infections. UPEC resistance to antibiotics decreases with children of the older age group. As well as this, molecular detections of ESBL-producing *E. coli* were more accurate than phenotypic identification. In addition, ESBL colonization and infection are increased in the urinary tract because of these genes’ products, *pai* and *hly*, as well as two β-lactamase genes, *bla_CTX_*_-M_ and *bla_TEM_*. The public health sectors would be better if they monitored the guidelines for the use of antibiotics and the careful use of antibiotics as recommended by physicians only. Furthermore, doctors need to give more advice about accurate hygiene, especially cleaning toddlers and pre-school children during toilet training to avoid UTIs. Hence, further molecular studies of different clinical specimens and different ages would achieve a confirmed database for ESBL-producing *E. coli* in Kurdistan, Iraq.

## Figures and Tables

**Figure 1 antibiotics-11-01246-f001:**
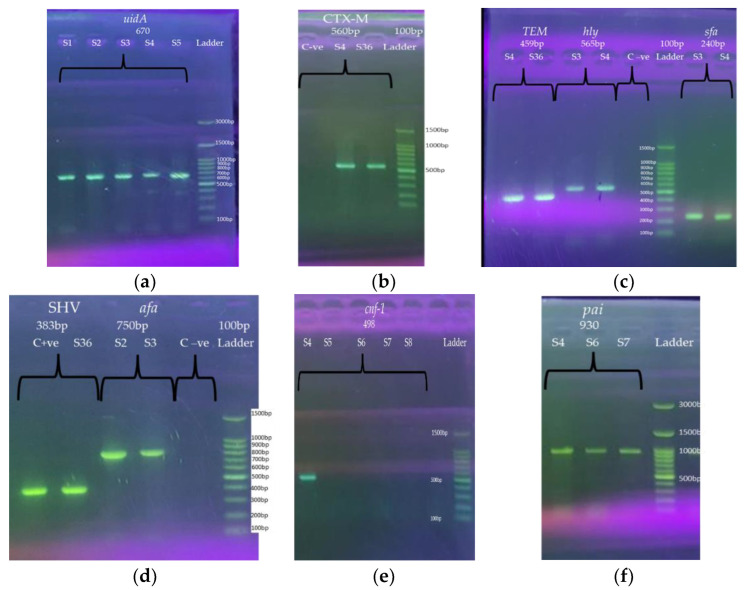
Gel electrophoresis of PCR products for *E. coli*-specific primers *uidA* (**a**) marker genes (*bla_TEM_, bla_CTX-M_*_,_ and *bla_SHV_*) (**b**–**d**) and five virulence factors marker genes (*pai, hly, cnf-1, sfa* and *afa*) (**c**–**f**). The amplified DNA fragments produce of various *E. coli* strains with these marker primers: species-specific primer *uidA* (lanes 1–5 for strains 1–5) (**a**), CTX-M (lanes 1, 2 and 3 for control negative and strains 4 and 36, respectively) (**b**), TEM (lanes 1 and 2 for strains 4 and 36), *hly* (lanes 3 and 4 for strains 3 and 4 and lane 5 for control negative), and *sfa* (lanes 7 and 8, for strains 3 and 4) (**c**), SHV (lanes 1 and 2, for strains control positive and S36), *afa* (lanes 3 and 4, for strains 2 and 3) (**d**), *cnf-1* (lanes 1, 2, 3, 4, 5 and 6 for strains 4, 5, 6, 7, 8 and control-ve) (**e**), and *pai* (lanes 1, 2, and 3, for strains 4, 6 and 7) (**f**), and lane 100 bp Ladder (GDSBio Marker) (**b**–**e**) and 300 bp (GeneDireX, Marker) as shown in (**a**,**f**). A 7 µL of the PCR products and ladder were pipetted into a prepared 1.5% agarose gel stained with 5 µL of Safe Gel Stain Dye. Key: *pai*; pathogenicity island; *hly*; hemolysin, *sfa*; S-fimbrial adhesion, *cnf-1*; cytotoxic necrotizing factor-1, and *afa*; a fimbrial adhesion and S = strain.

**Figure 2 antibiotics-11-01246-f002:**
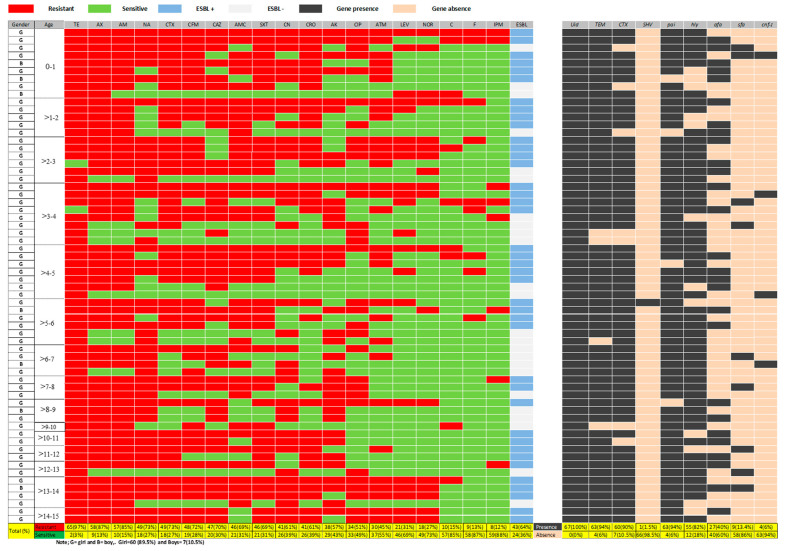
The prevalence of isolated *E. coli* and details of each strain for antibiotic resistance profiles and genes of both ESBL production and virulence factor. The red color of cells indicates that *E. coli* strains are resistant to antibiotics while the green color refers to *E. coli* strains that are sensitive to antibiotics. Furthermore, the blue color cells indicate that *E. coli* are ESBL producers by the DDST, whereas the white-darker color cells are for non-ESBL producer *E. coli*. Moreover, the black color cells indicate that *E. coli* strains have the marker genes while orange colors refer to *E. coli* strains which do not possess the marker. In addition, the yellow colors refer to the total and percentage for these above parameters. Each cell is for the individual bacteria. Keys: antibiotics include; TE: Tetracycline, AK: Amikacin, CAZ: Ceftazidime, AX: Amoxicillin, CTX: Cefotaxime, F: Nitrofurantoin, CN: Gentamicin, IPM: Imipenem, AM: Ampicillin, SXT: Trimethoprim/Sulphamethoxazole, CIP: Ciprofloxacin, LEV: Levofloxacin, AMC: Amoxicillin/Clavulanic acid, ATM: Aztreonam, CRO: Ceftriaxone, C: Chloramphenicol, CFM: Cefixime, NOR: Norfloxacin, NA: Nalidixic acid, and ESBL: ESBL-producing *E. coli.* Gene markers include *pai*: pathogenicity island, *hly*: hemolysin, *sfa*: S-fimbrial adhesion, *cnf-1*: cytotoxic necrotizing factor-1, and *afa*: a fimbrial adhesion, and β-lactamase genes: *bla_CTX_*_-M_, *bla_TEM_* and *bla_SHV_*.

**Figure 3 antibiotics-11-01246-f003:**
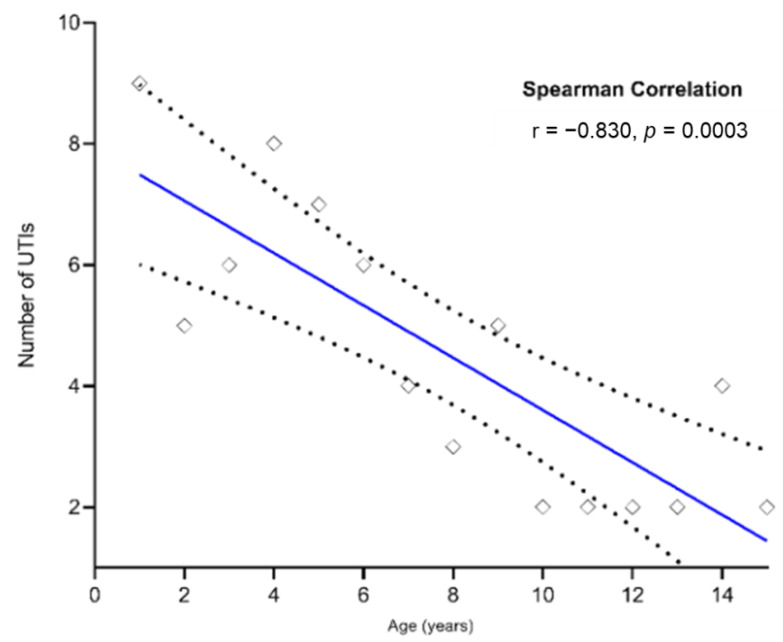
Correlation analysis between the children of all ages and UTIs by UPEC. The correlation analysis indicates that the number of UTIs reduces with children’s age, and hence this figure uses a number so that the relationship is evident across ages from 0 to 15 years. Each marker denotes a sum observation of UTIs for a particular age of the children. Spearman’s correlation was performed by GraphPad Prism, and significance was considered to be established when *p* < 0.05.

**Figure 4 antibiotics-11-01246-f004:**
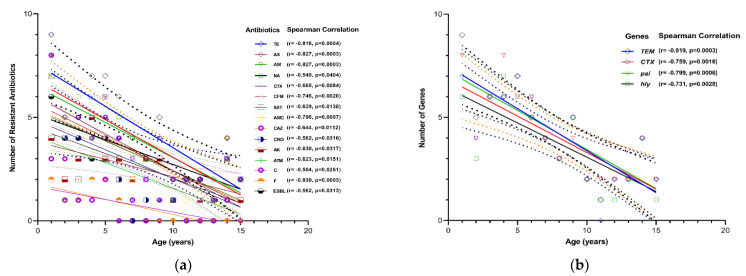
Correlation analysis between the children of all ages and antibiotic resistance to (**a**) and both the marker genes of ESBL-producing and virulence factors (**b**) of UPEC infection. This figure depicts the relationship between the antibiotic resistance to UPEC and the age of the children, where antibiotic resistance and gene markers for UPEC are illustrated in number terms. Each marker denotes a specific observation for antibiotics resistance and the age (years) of a particular child. This figure depicts the relationship between the UTIs with UPEC and the age of children, where UTIs are illustrated in number terms. Spearman’s correlation was performed by GraphPad Prism and a significance was considered to be established when *p* < 0.05. Spearman’s correlation method was carried out by GraphPad Prism, and significance was considered to be established when *p* < 0.05. Keys: TE: Tetracycline, AK: Amikacin, CAZ: Ceftazidime, AX: Amoxicillin, CTX: Cefotaxime, F: Nitrofurantoin, AM: Ampicillin, SXT: Trimethoprim/Sulphamethoxazole, AMC: Amoxicillin/Clavulanic acid, ATM: Aztreonam, CRO: Ceftriaxone, C: Chloramphenicol, CFM: Cefixime, NA: Nalidixic acid, and ESBL: ESBL-producing *E. coli*. Gene markers include *pai*: pathogenicity island, *hly*: hemolysin.

**Figure 5 antibiotics-11-01246-f005:**
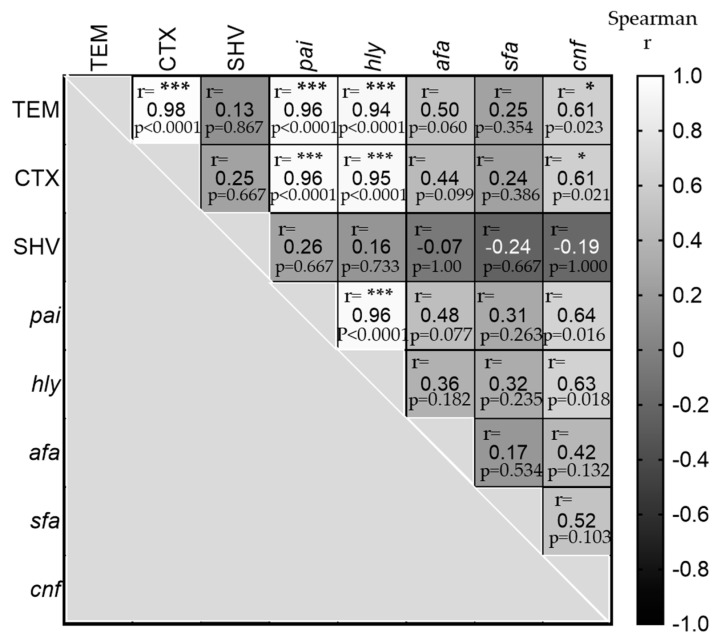
Correlation analysis among the antibiotic-resistant genes with virulence factors of UPEC infection. The heatmap shows the relationship among the antibiotic-resistant genes and virulence factors of UPEC infection, using computed nonparametric Spearman correlation. The white cells are highly positively correlated between these genes, r > 0.90 and significance were considered when the *p*-value was <0.0001 (***). At the same time, the light gray colors are moderate positively correlated, with r between 0.60–065 and a *p*-value < 0.05 (*). On the other hand, the darker gray cells are not significant found. Keys: Gene markers include *pai*: pathogenicity island, *hly*: hemolysin, *sfa*: S-fimbrial adhesion, *cnf-1*: cytotoxic necrotizing factor-1, and *afa*: a fimbrial adhesion.

**Table 1 antibiotics-11-01246-t001:** Representation of the antibiotic resistance of ESBL-producing *E. coli* and the virulence gene distribution pattern among of isolates UPEC.

ESBL-Producing *E. coli* Genes	No of Strains	Virulence Genes	No. of Strains
TEM, CTX-M, SHV	1	*pai, hly, afa, sfa*	2
TEM, CTX-M	58	*pai, hly, afa,*	18
TEM	4	*pai, hly, sfa, cnf-1*	1
CTX-M	1	*pai, hly*	21
		*pai, afa*	5
		*hly, afa*	2
		*pai, hly, sfa*	5
		*pai, hly, cnf-1*	3
		*pai, sfa*	1
		*pai*	5
		*hly*	1
		*afa*	1

**Table 2 antibiotics-11-01246-t002:** Primer names and sequences for different gene markers in this study.

Types	Genes	Oligonucleotide Sequence (5′–3′) Forward and Reverse	Amplicon Size (bp)	References
Species-specific gene	*uidA*	F:5′-CATTACGGCAAAGTGTGGGTCAAT-3′	658 bp	[81]
		R:5′-CCATCAGCACGTTATCGAATCCTT-3′		
Virulence genes	*pai*	F: 5′-GGACATCCTGGTACAGCGCGCA-3′	930 bp	[82]
		R: 5′-TCGCCACCAATCACAGCCGAAC-3′		
	*afa*	F: 5′-GCTGGGCAGCAAACTGATAACTCTC-3′	750 bp	[83]
		R: 5′-CATCAAGCTGTTTGTTCGTCCGCCG-3′		
	*hlyA*	F:5′-AGATTCTTGGGCATGTATCCT-3′	565 bp	[84]
		R:5′-TTGCTTTGCAGACTGTAGTGT-3′		
	*cnf-1*	F: 5′-AAGATGGAGTTTCCTATGCAGGAG-3′	498 bp	[81,85]
		R: 5′-CATTCAGAGTCCTGCCCTCATTATT-3′		
	*sfa*	F: 5′-GTGGATACGACGATTACTGTG- 3′	240 bp	[85]
		R: 5′-CCGCCAGCATTCCCTGTATTC-3′		
ESBL genes	*CTX-M*	F: 5′-GAAGGTCATCAAGAAGGTGCG-3′	560 bp	[86]
		R: 5′-GCATTGCCACGCTTTTCATAG-3′		
	*TEM*	F:5′-GAGACAATAACCCTGGTAAAT-3′	459 bp	
		R-5′-AGAAGTAAGTTGGCAGCAGTG-3′		
	*SHV*	F: 5′-GTCAGCGAAAAACACCTTGCC3′	383 bp	
		R: 5′-GTCTTATCGGCGATAAACCAG3′		

**Table 3 antibiotics-11-01246-t003:** The PCR amplification condition for detecting different gene markers in this study.

Genes	Initial Denaturation	Denaturation	Annealing	Extension	Final Extension	References
*uid*A	94 °C	92 °C	58 °C	72 °C	72 °C	[81]
	10 min	1 min	I min	30 s	5 min	
	1 cycle	35 cycles			1 cycle	
*afa*	94 °C	94 °C	63 °C	68 °C	72 °C	[83]
	5 min	1 min	1 min	3 min	7 min	
	1 cycle	30 cycles			1 cycle	
*sfa*	95 °C	94 °C	63 °C	68 °C	72 °C	[85]
	3 min	30 s	30 s	4 min	10 min	
	1 cycle	30 cycles			1 cycle	
*hly*	94 °C	94 °C	55 °C	72 °C	72 °C	[84]
	4 min	30 s	30 s	1 min	5 min	
	1 cycle	30 cycles			1 cycle	
*cnf-1*	95 °C	94 °C	68 °C	68 °C	72 °C	[81,85]
	3 min	30 sec	30 s	4 min	10 min	
	1 cycle	25 cycles			1 cycle	
*pai*	94 °C	94 °C	63 °C	72 °C	72 °C	[82,85]
	1 min	1 min	30 s	1.30 min	5 min	
	1 cycle	30 cycles			1 cycle	
TEM	94 °C	94 °C	55 °C	72 °C	72 °C	[86]
	3 min	45 s	30 sec	3 min	2 min	
	1 cycle	35 cycles			1 cycle	
SHV and CTX	94 °C	94 °C	60 °C	72 °C	72 °C	[86]
	3 min	45 s	30 s	3 min	2 min	
	1 cycle	35 cycles			1 cycle	

## Data Availability

All material information described in the manuscript and all relevant primary data, is openly accessible.

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
