# Peer review of "Molecular Characterization of Extended Spectrum β-Lactamase (ESBL) and Virulence Gene-Factors in Uropathogenic Escherichia coli (UPEC) in Children in Duhok City, Kurdistan Region, Iraq"

_antibiotics, 2022, doi:10.3390/antibiotics11091246_

Round 1

Reviewer 1 Report

Hassan and Ibrahim have reported the manuscript entitled as “Molecular characterization of Extended Spectrum β-lactamase (ESBL) with phenotypical detection and virulence gene-factors in uropathogenic Escherichia coli (UPEC) in children in Duhok Governorate, Kurdistan Region-Iraq” which actually is a synthesis regarding the antibiotic resistant E. coli prevalence in pediatric patients. The ms domain is very relevant and what-so-ever the data presented is concise and informative.

Major Points:

My 1st observation is the shortcoming of this ms for not to include the realtime PCR data that is quite common these days. The prevalence and frequencies of various genes (ARGs and virulence factors) would tell a lot in this study. Particularly when authors have somewhere talked about plasmid-born factors meaning the mobile genetic elements (MGEs) and their correlations would increase the quality and substance of this manuscript.

The authors have claimed to collect the samples from kids ‘not receiving antimicrobial treatment’; I just want to know the status of self-medication in Iraq that actually is quite a problem in developing countries. In that context, the claim might be quite more than the actual situation therein.

The authors have identified the E. coli isolates by biochemical tests; would not it be better to validate it with molecular tools as well e.g. 16s rRNA gene sequencing?

Figure-1: Please mentioned in legend what is a and b part of the figure. Moreover, the amplicon sizes of cnf (498), sfa (240bp) and pai (930bp) seem not to corroborate with the ladder provided here.

Figure-2: There are some discrepancies which authors need to see and correct e.g. in genes presence/absence data, what is the total number of sample? If it is same like biochemical test, then it should be 67 but it is either 71, 70, 124 etc. and similarly the percentage of presence/absence should cumulate to 100 that is not the case here….

Minors:

Please correct the ms linguistically via proof reading; some minor mistakes are mentioned below:

L14-15: change ‘is being conducted’ with “was conducted”

L22: Interestingly

L23: as well as

L24: change ‘illustrated is alarming increases’ with “illustrated alarming increase”

L26: change ‘sectors of public health should’ with “public health sectors should further”

L35: UTIs not UITs

L38: remove ‘in’ before febrile

L53-61: The paragraph is describing antibiotics and enzymes mechanisms and so the plasmid and plasmid coding genes do not suit here appropriately.

L60-61: The sentence starting with ‘Despite’ seems not complete.

L63: put full stop (.) after [22] not before.

L79: how 150 samples are 58% when the total numbers are 260?

L110: another isolate not isolated…

L310: was more accurate not accuracy

L317/L319: gene markers not genes markers

Author Response

Dear Dr Karina Yang,

Manuscript ID antibiotics-1879787 entitled " Molecular characterization of Extended Spectrum β-lactamase (ESBL) with phenotypical detection and virulence gene-factors in uropathogenic Escherichia coli (UPEC) in children in Duhok Governorate, Kurdistan Region-Iraq".

We thank your and the reviewers for the work put into consideration of our submission and we are very happy to revise the manuscript in line with the recommendations.

We indicate the colour changes made in text background.

Yours sincerely

Dr K Ibrahim

28/08/2022

////////////////////////////////////////////////////////////////////////

Author's Reply to the Review Report (Reviewer 1)

Comments and Suggestions for Authors

Hassan and Ibrahim have reported the manuscript entitled as “Molecular characterization of Extended Spectrum β-lactamase (ESBL) with phenotypical detection and virulence gene-factors in uropathogenic Escherichia coli (UPEC) in children in Duhok Governorate, Kurdistan Region-Iraq” which actually is a synthesis regarding the antibiotic resistant E. coli prevalence in pediatric patients. The ms domain is very relevant and what-so-ever the data presented is concise and informative.

Major Points:

My 1st observation is the shortcoming of this ms for not to include the realtime PCR data that is quite common these days. The prevalence and frequencies of various genes (ARGs and virulence factors) would tell a lot in this study. Particularly when authors have somewhere talked about plasmid-born factors meaning the mobile genetic elements (MGEs) and their correlations would increase the quality and substance of this manuscript.

  • Has been removed the “Plasmids usually carry genes encoding resistance to other drug classes, such as aminoglycosides, and these plasmids are responsible for producing ESBLs, which are frequently plasmid-encoded [20,21]

The authors have claimed to collect the samples from kids ‘not receiving antimicrobial treatment’; I just want to know the status of self-medication in Iraq that actually is quite a problem in developing countries. In that context, the claim might be quite more than the actual situation therein.

We have modified the conclusion

  • In Kurdistan-Iraq, there is not like western countries for having Electronic Medical record. Thus, any Investigator has the questioner (takes case history from the mother) according to the Ethical forum,

The authors have identified the E. coli isolates by biochemical tests; would not it be better to validate it with molecular tools as well e.g. 16s rRNA gene sequencing?

We have modified the conclusion

  • In this study we identified the E. coli not only by biochemical tests; but also the species-specific primers, UidA gene, and also 3 primers to detect ESBL; blaCTX-M, blaTEM and blaSHV β-lactamase genes.
  • The limitation of this study is the lack of molecular characterization, 16s rRNA gene sequencing, and whole genome sequencing of E. coli for the phylogenetic analysis might help understand the epidemiology and clinical microbial infection of UPEC in our community.
  •  

Figure-1: Please mentioned in legend what is a and b part of the figure. Moreover, the amplicon sizes of cnf (498), sfa (240bp) and pai (930bp) seem not to corroborate with the ladder provided here.

  • Has been modified and updated almost result in the text, figures title and legend as well
  • In the text L 95 – 121
  • In Figure1 change to Figure 1a, b, c, d, e
  • The legend and keys also changed 164-178

Figure-2: There are some discrepancies which authors need to see and correct e.g. in genes presence/absence data, what is the total number of sample? If it is same like biochemical test, then it should be 67 but it is either 71, 70, 124 etc. and similarly the percentage of presence/absence should cumulate to 100 that is not the case here….

  • Thank you so much for your comments and it was math issues
  • It has been modified, and changed and the necessary update in the text and in Figure 2 and it is legend (The yellow colour we are representing both number and % of each antibiotic and marker genes and)
  • In the text L 85-121
  • In the legend and keys 180-196

Minors:

Please correct the ms linguistically via proof reading; some minor mistakes are mentioned below:

L14-15: change ‘is being conducted’ with “was conducted”

  • Has been changed. Now L 13-14

L22: Interestingly

  • Has been changed. Now L 21

L23: as well as

  • Has been changed. Now L 23

L24: change ‘illustrated is alarming increases’ with “illustrated alarming increase”

  • Has been changed. Now L 22

L26: change ‘sectors of public health should’ with “public health sectors should further”

  • Has been changed. Now L 24

L35: UTIs not UITs

  • Has been changed with update the sentence. Now L 32

L38: remove ‘in’ before febrile

  • Has been removed with update the sentences. L31-39

L53-61: The paragraph is describing antibiotics and enzymes mechanisms and so the plasmid and plasmid coding genes do not suit here appropriately.

  • Has been removed the “Plasmids usually carry genes encoding resistance to other drug classes, such as aminoglycosides, and these plasmids are responsible for producing ESBLs, which are frequently plasmid-encoded [20,21]

L60-61: The sentence starting with ‘Despite’ seems not complete.

  • Sentence has been changed and added 2 another references
  • For serious infections caused by ESBL-producing organisms, carbapenems are the drug of choice [21]. However, studies reported that carbapenem-resistant isolates do exist [21,22]
  • (Paterson, D.L., 2000) and (Legese et al 2017)

L63: put full stop (.) after [22] not before.

  • Has been removed.

L79: how 150 samples are 58% when the total numbers are 260?

  • Has been modified, and changed in the necessary update in the text.
  • From these cultures 150 (57.7%) L76
  • 67 (44.7%) L78

L110: another isolate not isolated…

  • Has been removed and updated the paragraph. L 94-107

L310: was more accurate not accuracy

  • Has been removed. L 389

L317/L319: gene markers not genes markers

  • Has been changed.

Table 2. Primer names and sequences for different gene markers in this study L 401

Table 3. The PCR amplification condition for detecting different gene markers in this study. L 404

Reviewer 2 Report

The authors in this manuscript have focused on molecular characterization of the UPEC strains isolated from children with clinical suspicion of UTI. Using gene specific primers for virulence factors and antibiotic resistance markers, the authors have pointed out the alarming emergence of multi drug resistance in children and have found correlation with the age as well. Overall, the study undertaken by the authors is relevant and significant to the clinical setting given that findings like these help the public health sector in revising/optimizing guidelines for the the optimal use of antibiotics.

Following are my major comments:

1.  In figure 1, please make clear what is the diffrence between lane 1 Vs 2, 3 Vs 4 and so on. It is neither clear in the figure legend nor in the methodology as to what are the two different DNA samples that are being analyzed for the mentioned genes. In line 134 what do you refer to as "two various samples?"

2. Results from the age correlation studies need to be rewritten. Graphs in figure 3 aren't as self explanatory. A more elaborate figure legend and well explained  text is needed to bring the reader's attention to the conclusion.

Minor comments:

Line 35: UTI, instead of UIT

Line 56: ineffective, instead of ineffectual

Line 61: do exist, instead of to exist

Line 90: which showed/depicted, instead of which found

Line 105: interesting, instead of interested

Line 149: predominant, instead of predominate

Line 188: resistant to, instead of resistance to

Line 308: increase, instead of increases

Line 310: more accurate than phenotypic, instead of more accuracy than phenotypically

Author Response

Dear Dr Karina Yang,

Manuscript ID antibiotics-1879787 entitled " Molecular characterization of Extended Spectrum β-lactamase (ESBL) with phenotypical detection and virulence gene-factors in uropathogenic Escherichia coli (UPEC) in children in Duhok Governorate, Kurdistan Region-Iraq".

We thank your and the reviewers for the work put into consideration of our submission and we are very happy to revise the manuscript in line with the recommendations.

We indicate the colour changes made in text background.

Yours sincerely

Dr K Ibrahim

28/08/2022

////////////////////////////////////////////////////////////////////////

Author's Reply to the Review Report (Reviewer 2)

Comments and Suggestions for Authors

The authors in this manuscript have focused on molecular characterization of the UPEC strains isolated from children with clinical suspicion of UTI. Using gene specific primers for virulence factors and antibiotic resistance markers, the authors have pointed out the alarming emergence of multi drug resistance in children and have found correlation with the age as well. Overall, the study undertaken by the authors is relevant and significant to the clinical setting given that findings like these help the public health sector in revising/optimizing guidelines for the the optimal use of antibiotics.

Following are my major comments:

  1. In figure 1, please make clear what is the diffrence between lane 1 Vs 2, 3 Vs 4 and so on. It is neither clear in the figure legend nor in the methodology as to what are the two different DNA samples that are being analyzed for the mentioned genes. In line 134 what do you refer to as "two various samples?"
  • Has been changed and updated. As Reviewer 1 recommended as well.
  • Has been modified and updated almost result in the text, figures title and legend as well
  • In the text L 95 – 121
  • In Figure1 change to Figure 1a, b, c, d, e
  • The legend and keys also changed 164-178
  1. Results from the age correlation studies need to be rewritten. Graphs in figure 3 aren't as self explanatory. A more elaborate figure legend and well explained  text is needed to bring the reader's attention to the conclusion.
  • Has been changed and updated as also recommend the reviewer 3 too
  • Figure 3 changed to two figures; Figure 3 and Figure 4a&b modified and changed the necessary appropriate in the text
  • In the Text L 123- 159
  • In figures titles and legends L198- 220

Minor comments:

Line 35: UTI, instead of UIT

  • Has been changed with update the sentence. Now L 32

Line 56: ineffective, instead of ineffectual

  • Has been changed. L 55

Line 61: do exist, instead of to exist

  • Sentence has been changed and added 2 another references
  • For serious infections caused by ESBL-producing organisms, carbapenems are the drug of choice [21]. However, studies reported that carbapenem-resistant isolates do exist [21,22]
  • (Paterson, D.L., 2000) and (Legese et al 2017)

Line 90: which showed/depicted, instead of which found

  • Several sentences have been changed. L 89-95

High antibiotic resistance rates were detected in the E. coli which found that they were almost resistant to Tetracycline. Approximately 85% of isolated E. coli were highly resistant to Amoxicillin and Ampicillin. Furthermore, around 70% of isolated E. coli were resistant to Nalidixic acid, Cefotaxime, Cefixime, Trimethoprim/Sulphamethoxazole, Gentamicin and Ceftriaxone, whereas around 50% of isolated E. coli were moderate resistant to Amikacin, Ciprofloxacin and Aztreonam. In contrast, over 85% of E. coli were highly sensitive to Chloramphenicol, Nitrofurantoin and Imipenem.

change to ; lines 85-91

All isolated E. coli were resistant to Tetracycline (100%) and approximately 85% of them were highly resistant to Amoxicillin and Ampicillin. Furthermore, around 70% of isolated E. coli were resistant to Nalidixic acid, Cefotaxime, Cefixime, Trimethoprim/ Sulphamethoxazole, Gentamicin and Ceftriaxone. In addition, around (55%) of these E. coli were resistant to Amikacin, (49%) to Ciprofloxacin and (46%) to Aztreonam. In contrast, over 85% of E. coli were highly susceptible to Chloramphenicol, Nitrofuranto-in and Imipenem

Line 105: interesting, instead of interested

  • Has been changed L 102
  • Almost sentences have been changed and update

Line 149: predominant, instead of predominate

  • Has been changed. L 227
  •  

Line 188: resistant to, instead of resistance to

  • Has been changed. L 272
  •  

Line 308: increase, instead of increases

  • Has been changed. L 385
  •  

Line 310: more accurate than phenotypic, instead of more accuracy than phenotypically

  • We have also modified the conclusion paragraph
  • increases change to increase. L 391

Reviewer 3 Report

Title; Title should be concise.

Abstract: Line 22. Remove Phenotypically.

Introduction:

Line 34-35. The total US healthcare costs 34 for managing and treating a UTI was $630 million in 2013

Add recent data…

Line 35-37.  Add recent data.

Results

Line 105-107….Sentence is too lengthy. Concise it

Line 13-14. Remove gene

Line 14-19. Sentences are too lengthy. Revise these sentences and describe these result in short sentences.

Results: What do you mean by samples?  As you mentioned “The three virulence genes of pai, hly and sfa genes were found in only 5 of the 120 isolated E. coli samples” I think it is E. coli strain/instead of E. coli samples.

Line: 121. Italicize the pai.

Line 16-18. Sentence is very complicated and difficult to understand the actual observation. Revise this.

In overall result section sentences are too lengthy.

Discussion:

Author should focus on his own findings instead of comparing other regional studies. The authors have mentioned that previous studies have raised this issue in same region of Iraq. Why this study was carried. Author should mention the uniqueness of this study.

Conclusion: Must be revised and focus on findings of the study.  

Author Response

Dear Dr Karina Yang,

Manuscript ID antibiotics-1879787 entitled " Molecular characterization of Extended Spectrum β-lactamase (ESBL) with phenotypical detection and virulence gene-factors in uropathogenic Escherichia coli (UPEC) in children in Duhok Governorate, Kurdistan Region-Iraq".

We thank your and the reviewers for the work put into consideration of our submission and we are very happy to revise the manuscript in line with the recommendations.

We indicate the colour changes made in text background.

Yours sincerely

Dr K Ibrahim

28/08/2022

////////////////////////////////////////////////////////////////////////

Author's Reply to the Review Report (Reviewer 3)

Comments and Suggestions for Authors

Title; Title should be concise.

Molecular characterization of Extended Spectrum β-lactamase (ESBL) with phenotypical detection and virulence gene-factors in uropathogenic Escherichia coli (UPEC) in children in Duhok Governorate, Kurdistan Region-Iraq

  • Has been changed to

Molecular characterization of Extended Spectrum β-lactamase (ESBL) and virulence gene-factors in uropathogenic Escherichia coli (UPEC) in children in Duhok City, Kurdistan Region-Iraq

Abstract: Line 22. Remove Phenotypically.

  • Has been removed.

Introduction:

Line 34-35. The total US healthcare costs 34 for managing and treating a UTI was $630 million in 2013   Add recent data…   Line 35-37.  Add recent data.

In USA, paediatric UTIs are estimated to impact 2.4 to 3 % of all American children each year, resulting in more than 1.5 million doctor visits [5] and the total US healthcare costs for managing and treating a UTI was $630 million in 2013 [6]. Annually, UITs accounted for 0.7% of visits to physician offices and between 5% and 14% of visits to emergency departments by children [7]. The incidence of UTIs is common during childhood, and affect equally in both boys and girls during the first year of life [8]. In febrile infants aged 2-3 months were documented 3% to 10% with urinary symptoms [8] and after that age before 7 years, girls account for the majority of infections approximately 8.4% and less for 1.7% of boys [9–11].

  • Has been changed and updated. Added another reference (Flores-Mireles et al 2015)

On yearly basis, the total healthcare cost of UTIs in the USA, whether for its management or treatment, is over $ 3.5 billion (Flores-Mireles  et al 2015), and around $630 million for paediatrics UTIs [6]. Paediatric UTIs are estimated to impact 2.4 to 3 % of all American children each year, resulting in more than 1.5 million doctor visits [5], of which 5% - 14% dedicated to emergency visits [7]. The incidence of UTIs is common during childhood, and affect equally in both boys and girls during the first year of life [8]. UTIs are considered the second most frequent bacterial infection in children throughout the first seven years of life and girls account for the majority of infections, approximately 8.4% and less than 1.7% for boys [9–11].

Results

Line 105-107….Sentence is too lengthy. Concise it

Line 13-14. Remove gene

Line 14-19. Sentences are too lengthy. Revise these sentences and describe these result in short sentences.

Results: What do you mean by samples?  As you mentioned “The three virulence genes of pai, hly and sfa genes were found in only 5 of the 120 isolated E. coli samples” I think it is E. coli strain/instead of E. coli samples.

  • Has been changed (sample to strain) in the necessary update in text

Line: 121. Italicize the pai.

  • Has been changed to

Line 16-18. Sentence is very complicated and difficult to understand the actual observation. Revise this.

In overall result section sentences are too lengthy.

  • Has been changed and updated almost in the results section including legends
  • In text L85-159
  • All legend has been modified and updated

Discussion:

Author should focus on his own findings instead of comparing other regional studies. The authors have mentioned that previous studies have raised this issue in same region of Iraq. Why this study was carried. Author should mention the uniqueness of this study.

  • Has been mentioned in the introduction Line 67-69

In Kurdistan Region-Iraq, several studies have been conducted on the prevalence of UTIs among adults [32–34] and reporting only the antibiotics susceptibility test in children [35–37]

  • Has been changed and update the last paragraph L 306-309

Regarding the age, it is noteworthy in this study that the negative correlation between the age of paediatrics and UTIs by UPEC with their resistant to antibiotics and both genes of resistance and virulence factors. This finding has not been previously reported in Iraq

Conclusion: Must be revised and focus on findings of the study.  

In conclusion, the data illustrate alarming increases of E. coli with ESBL-production in UTIs in early age children and found higher in girls than boys. Molecular detection of ESBL-producing E. coli was more accuracy than phenotypically identification. Further molecular studies of different clinical specimens and different age would achieve a confirmed database for ESBL-producing E. coli in Kurdistan, Iraq. In addition, the sectors of public health should monitor guidelines for using antibiotics.

  • We have revised, changed and added sentences to conclusion 388-40

Reviewer 4 Report

Overall, this manuscript has scientific merit that warrants publication in Antibiotics. However, the manuscript is difficult to read because it was poorly written. Therefore, the manuscript should be extensively revised prior to publication. The most important thing is for the authors to get a native English speaker to extensively revise the English language of this manuscript.

Line 38 to 41: The message that the authors attempted to convey in this sentence is not clear. Please revise the sentence.

Line 60 to 61: Please revise the sentence because it is not clear.

Line 63: Grammatical error. Please correct the placement of 'dot' after 'parenthesis'. That is '[23].'

Line 89 to 91: Please revise the sentence.

Line 94 to 95: "whereas around 50% of isolated E. coli were moderate resistant to Amikacin, Ciprofloxacin and Aztreonam." Please revise this sentence.

Line 105: "It is interested" change to "It is interesting"

Line 105 to 107: Please revise the sentence. The message is not clear.

Line 109 to 111: Please revise the sentence.

Line 114 to 117: Please revise the sentence.

Line 120 to 121: Please revise the sentence.

Line 126 to 128: Please revise the sentence.

Line 126 to 128: Although the authors already provided P values in Figure 3, It will be more interesting if the authors can provide P value for data analysis in the text. 

Besides, Figure 3 is difficult to read. Perhaps the authors can try to revise Figure 3 to make it more presentable.

Line 149: "Predominate" or "predominant"?

Line 174: "Tetracycline" (first letter uppercase) or "tetracycline (first letter lower case)" ? Please check.

Also I found the authors capitalized other antibiotic names such as Ampicillin, Amoxicillin, and Penicillin. Please check whether antibiotic names need to be capitalized or not. 

Line 187 to 189: Please revise the sentence.

Line 212 to 214: Please revise the sentence.

Line 214 to 216: Can you explain more why 25% missed phenotypically? Is there any study that found similar finding?

Line 275: How do you check the quality of DNA from gene amplification? Usually we check the quality of DNA using Nanodrop or gel. I can't find any relevant information from Table 2. Please revise or elaborate how do you check quality of DNA.

Line 287:  Seems like there are two Table 2. Please check because I found two Table 2 in the manuscript. Also, the description of Table 2 and Table 3 in the text does not consistent with what is presented in Table 2 and Table 3. Please revise the sentence and make sure the Table description in text consistent with what was presented in Table 2 and 3.

Line 310; Please change "accuracy" to "accurate".

Author Response

Dear Dr Karina Yang,

Manuscript ID antibiotics-1879787 entitled " Molecular characterization of Extended Spectrum β-lactamase (ESBL) with phenotypical detection and virulence gene-factors in uropathogenic Escherichia coli (UPEC) in children in Duhok Governorate, Kurdistan Region-Iraq".

We thank your and the reviewers for the work put into consideration of our submission and we are very happy to revise the manuscript in line with the recommendations.

We indicate the colour changes made in text background.

Yours sincerely

Dr K Ibrahim

28/08/2022

////////////////////////////////////////////////////////////////////////

Author's Reply to the Review Report (Reviewer 4)

Comments and Suggestions for Authors

Overall, this manuscript has scientific merit that warrants publication in Antibiotics. However, the manuscript is difficult to read because it was poorly written. Therefore, the manuscript should be extensively revised prior to publication. The most important thing is for the authors to get a native English speaker to extensively revise the English language of this manuscript.

Line 38 to 41: The message that the authors attempted to convey in this sentence is not clear. Please revise the sentence.

In USA, paediatric UTIs are estimated to impact 2.4 to 3 % of all American children each year, resulting in more than 1.5 million doctor visits [5] and the total US healthcare costs for managing and treating a UTI was $630 million in 2013 [6]. Annually, UITs accounted for 0.7% of visits to physician offices and between 5% and 14% of visits to emergency departments by children [7]. The incidence of UTIs is common during childhood, and affect equally in both boys and girls during the first year of life [8]. In febrile infants aged 2-3 months were documented 3% to 10% with urinary symptoms [8] and after that age before 7 years, girls account for the majority of infections approximately 8.4% and less for 1.7% of boys [9–11].

  • Has been changed and updated. Added another reference (Flores-Mireles et al 2015)
  • Also, reviewer 3 recommended

On yearly basis, the total healthcare cost of UTIs in the USA, whether for its management or treatment, is over $ 3.5 billion (Flores-Mireles  et al 2015), and around $630 million for paediatrics UTIs [6]. Paediatric UTIs are estimated to impact 2.4 to 3 % of all American children each year, resulting in more than 1.5 million doctor visits [5], of which 5% - 14% dedicated to emergency visits [7]. The incidence of UTIs is common during childhood, and affect equally in both boys and girls during the first year of life [8]. UTIs are considered the second most frequent bacterial infection in children throughout the first seven years of life and girls account for the majority of infections, approximately 8.4% and less than 1.7% for boys [9–11].

Line 60 to 61: Please revise the sentence because it is not clear.

Despite this, previously reported carbapenem-resistant isolates to exist [20]

  • Reviewer 1 also recommended
  • Sentence has been changed and revised as well as added 2 another references

For serious infections caused by ESBL-producing organisms, carbapenems are the drug of choice [21]. However,  studies reported that carbapenem-resistant isolates do exist [21,22]

(Paterson, D.L., 2000) and (Legese et al 2017)

Line 63: Grammatical error. Please correct the placement of 'dot' after 'parenthesis'. That is '[23].'

  • Has been removed.

Line 89 to 91: Please revise the sentence.

Line 94 to 95: "whereas around 50% of isolated E. coli were moderate resistant to Amikacin, Ciprofloxacin and Aztreonam." Please revise this sentence.

Line 105: "It is interested" change to "It is interesting"

Line 105 to 107: Please revise the sentence. The message is not clear.

Line 109 to 111: Please revise the sentence.

Line 114 to 117: Please revise the sentence.

Line 120 to 121: Please revise the sentence.

Line 126 to 128: Please revise the sentence.

Line 126 to 128: Although the authors already provided P values in Figure 3, It will be more interesting if the authors can provide P value for data analysis in the text. 

Besides, Figure 3 is difficult to read. Perhaps the authors can try to revise Figure 3 to make it more presentable.

  • In General the results have been modified, changed and updated the necessary ( Reviewer 1 and 3 also recommended).
  • Has been modified and updated almost result in the text, figures title and legend as well
  • P-value also added to text and legend
  • In the text L 95 – 121
  • In Figure1 change to Figure 1a, b, c, d, e
  • The legend and keys also changed 164-178

  • Has been changed and updated ( also recommend by the reviewer 3)
  • Figure 3 changed to two figures; Figure 3 and Figure 4a&b modified and changed the necessary appropriate in the text
  • In the Text L 123- 159
  • In figures titles and legends L198- 220

Line 149: "Predominate" or "predominant"?

  • Has been changed. L227

Line 174: "Tetracycline" (first letter uppercase) or "tetracycline (first letter lower case)" ? Please check.

Also I found the authors capitalized other antibiotic names such as Ampicillin, Amoxicillin, and Penicillin. Please check whether antibiotic names need to be capitalized or not. 

  • Has been changed and updated.

Line 21, amoxicillin, ampicillin and tetracycline (Amoxicillin, Ampicillin and Tetracycline)

Line 174 tetracycline changed Tetracycline

Line TE; Tetracycline, AK; Amikacin

Line 187 to 189: Please revise the sentence.

Several previous studies also reported that E. coli have not been observed resistance to Imipenem of UTIs patients under 12 old in Iran [55], and in the various ages in Saudi Arabia [56] as well as in Iraqi studies; Duhok city [57] and Erbil city [58].

  • We have modified the sentence and changed the necessary update. L266-277

Line 212 to 214: Please revise the sentence.

Although the phenotypically identified ESBL-producing E. coli was 64%, un expectable of the 94% and 89% of all the E. coli were harbored for ESBL genes, TEM and CTX-M, respectively.

  • We have revised and modified the sentences and changed the necessary update. L287-297

Line 214 to 216: Can you explain more why 25% missed phenotypically? Is there any study that found similar finding?

However, they were negative phenotypically; about 85% were harboring ESBL genes, which means that about 25% missed phenotypically.

  • We have revised and modified the sentences and changed the necessary update. L287-297

Line 275: How do you check the quality of DNA from gene amplification? Usually we check the quality of DNA using Nanodrop or gel. I can't find any relevant information from Table 2. Please revise or elaborate how do you check quality of DNA.

  • Has been changed.
  • We have revised and modified the sentences. L356-358

Bacterial DNA quality was achieved by (NanoDrop™ One UV-Vis Spectrophotometer, Thermo Fisher Scientific, USA)

Line 287:  Seems like there are two Table 2. Please check because I found two Table 2 in the manuscript. Also, the description of Table 2 and Table 3 in the text does not consistent with what is presented in Table 2 and Table 3. Please revise the sentence and make sure the Table description in text consistent with what was presented in Table 2 and 3.

  • We have revised and modified.

Line 310; Please change "accuracy" to "accurate".

  • Has been changed.  L39

Round 2

Reviewer 1 Report

Hassan and Ibrahim have reported the manuscript entitled as “Molecular characterization of Extended Spectrum β-lactamase (ESBL) with phenotypical detection and virulence gene-factors in uropathogenic Escherichia coli (UPEC) in children in Duhok Governorate, Kurdistan Region-Iraq”. The revised manuscript is modified according to earlier comments but the main points are still there e.g.

What is authors’ response for not to include the realtime PCR data in the ms. The prevalence and frequencies of various genes (ARGs and virulence factors) would tell a lot in this interesting work.

I am still curious for authors’ observation/record on the status of self-medication in Iraq that actually is quite a problem in developing countries.

The authors have not mentioned anything regarding the identification of their E. coli isolates with molecular tools like 16s rRNA gene sequencing?

Figure-2: Again, for SHV, 1-present and 67-absent mean the total of 68 while the total in all other cases is 67. Authors need serious attention to present their data.

Author Response

We thank your and the reviewers for the work put into consideration of our submission and we are very happy to revise the manuscript in line with the recommendations.

We indicate the colour changes made in text background.

Yours sincerely

Dr K Ibrahim

////////////////////////////////////////////////////////////////////////

Author's Reply to the Review Report (Reviewer 1)

 Comments and Suggestions for Authors

Hassan and Ibrahim have reported the manuscript entitled as “Molecular characterization of Extended Spectrum β-lactamase (ESBL) with phenotypical detection and virulence gene-factors in uropathogenic Escherichia coli (UPEC) in children in Duhok Governorate, Kurdistan Region-Iraq”. The revised manuscript is modified according to earlier comments but the main points are still there e.g.

What is authors’ response for not to include the realtime PCR data in the ms. The prevalence and frequencies of various genes (ARGs and virulence factors) would tell a lot in this interesting work.

  • Has been added in result a subtitle and figure 5 with legend L 219-238

2.6 Relationship between antibiotics resistance and virulence factors genes

Figure (5) illustrates the relationship betweenthe antibiotic-resistant genes and virulence factors of UPEC infection, using Spearman’s correlation coefficient. There were highly significant positive correlations between TEM and CTX (r= 0.98, p-value= 0.0000), pai (r=0.96, p-value= 0.000) and hly (r=0.94, p-value= 0.000), and also between CTX, and pai (r= 0.96, p-value= 0.0000) and hly (r=0.95, p-value= 0.000). Furthermore, although the very low prevalence of cnf, it also found that cnf was a significant positive correlation with TEM (r=0.60, p-value= 0.023), CTX (r=0.60, p-value= 0.021), pai (r=0.63, p-value= 0.016), and hly (r=0.62, p-value= 0.018).

Figure 5. Correlation analysis among the antibiotic- resistant genes with virulence factors of UPEC infection.

The heatmap shows the relationship among the antibiotic-resistant genes and virulence factors of UPEC infection, using compute nonparametric Spearman correlation. The white cells are highly positive correlated between these genes, r>0.90 and significance were considered when the p-value was > 0.00001. At the same time the light grey colors are moderate positively correlated, with r between 0.60-065 and a p-value>0.05. On the other hand, the darker grey cells are not significant found.  Keys: Gene markers include pai; pathogenicity island; hly; hemolysin, sfa; S-fimbrial adhesion, cnf-1; cytotoxic necrotizing factor-1, and afa; a fimbrial adhesion.

  • Has been added in a paragraph in discussion as well L 331-336

In addition, further analysis was measured to answer whether the virulence genes have or not a role of increase antibiotics resistant genes. Only one study was reported in Egypt by Abd El-Baky et al [72] and found a significant positive correlation between CTX and hly which it is similar to this study. Thus, further molecular studies of whole genome sequencing are recommended to highlight the relation between the virulence factors and antibiotic-resistant genes of these pathogenic E.coli.

I am still curious for authors’ observation/record on the status of self-medication in Iraq that actually is quite a problem in developing countries.

  • In my country is very easy to buy antibiotics in any time from pharmacy and small nurse clinic without prescriptions. Thus, in this case sometime I am calling “hazelnuts – biotic)

The authors have not mentioned anything regarding the identification of their E. coli isolates with molecular tools like 16s rRNA gene sequencing?

  • Has been added 2 sentences with 4 references in the discussion L311-313

In this study all isolated E. coli were confirmed by the PCR amplification of E. coli-specific marker gene (uidA). At it is known, this gene is widely used in identification of E. coli from clinical samples [63–66].

  1. Cleuziat, P.; Robert-Baudouy, J. Specific Detection of Escherichia Coli and Shigella Species Using Fragments of Genes Coding for β-Glucuronidase. FEMS Microbiol. Lett. 1990, 72, 315–322.
  2. Feng, P.; Lum, R.; Chang, G.W. Identification of UidA Gene Sequences in Beta-D-Glucuronidase-Negative Escherichia Coli. Appl. Environ. Microbiol. 1991, 57, 320–323.
  3. Farnleitner, A.H.; Kreuzinger, N.; Kavka, G.G.; Grillenberger, S.; Rath, J.; Mach, R.L. Simultaneous Detection and Differentiation of Escherichia Coli Populations from Environmental Freshwaters by Means of Sequence Variations in a Fragment of the β-D-Glucuronidase Gene. Appl. Environ. Microbiol. 2000, 66, 1340–1346.
  4. Brons, J.K.; Vink, S.N.; de Vos, M.G.J.; Reuter, S.; Dobrindt, U.; van Elsas, J.D. Fast Identification of Escherichia Coli in Urinary Tract Infections Using a Virulence Gene Based PCR Approach in a Novel Thermal Cycler. J. Microbiol. Methods 2020, 169, 105799.

Figure-2: Again, for SHV, 1-present and 67-absent mean the total of 68 while the total in all other cases is 67. Authors need serious attention to present their data.

  • Has been modified and updated figure 2 (SHV presence 1 (1.5%) and absence 66(98.5%)

Reviewer 3 Report

The author should focus on Mechanism instead  of prevalence. Some how author should avoid redundancy. otherwise revision is looks well in its present version. 

Author Response

We thank your and the reviewers for the work put into consideration of our submission and we are very happy to revise the manuscript in line with the recommendations.

We indicate the colour changes made in text background.

Yours sincerely

Dr K Ibrahim

////////////////////////////////////////////////////////////////////////

Author's Reply to the Review Report (Reviewer 3)

The author should focus on Mechanism instead  of prevalence. Some how author should avoid redundancy. otherwise revision is looks well in its present version. 

  • Has been added some sentences with references in discussion L313 -320

Evidence demonstrated that the presence of pai in pathogenic strains carry genes encoding one or more virulence factors, such as adhesins, toxins (cnf1 and hly), invasions, and iron absorption systems [67,68]. Altogether, these bacterial virulence factors enhance their ability to adhere, invade and colonize a host and increase the pathogenicity [69,70]. Besides, these virulence factors also help pathogenic bacteria to become resistant to immune defenses, like phagocytosis, the complement system, or adaptive immune responses [69–71]. Altogether these genes factors in UPEC contribute to urinary virulence associated with severe UTIs [18].

  1. Tarchouna, M.; Ferjani, A.; Ben-Selma, W.; Boukadida, J. Distribution of Uropathogenic Virulence Genes in Escherichia Coli Isolated from Patients with Urinary Tract Infection. Int. J. Infect. Dis. 2013, 17, 10–13, doi:10.1016/j.ijid.2013.01.025.
  2. Schroeder, M.; Brooks, B.D.; Brooks, A.E. The Complex Relationship between Virulence and Antibiotic Resistance. Genes (Basel). 2017, 8, 39.
  3. Giaouris, E.; Heir, E.; Desvaux, M.; Hébraud, M.; Møretrø, T.; Langsrud, S.; Doulgeraki, A.; Nychas, G.-J.; Kačániová, M.; Czaczyk, K. Intra-and Inter-Species Interactions within Biofilms of Important Foodborne Bacterial Pathogens. Front. Microbiol. 2015, 6, 841.
  4. Srivastava, N.; Srivastava, M.; Mishra, P.K.; Ramteke, P.W.; Singh, R.L. New and Future Developments in Microbial Biotechnology and Bioengineering: From Cellulose to Cellulase: Strategies to Improve Biofuel Production; Elsevier, 2019; ISBN

  • Thank you so much for your previous comments

Reviewer 4 Report

All questions and suggestions have been answered. No more comment/question from me.

Author Response

Author's Reply to the Review Report (Reviewer 4)

Comments and Suggestions for Authors

All questions and suggestions have been answered. No more comment/question from me.

  • Thank you so much for your previous comments